# A genetic locus complements resistance to *Bordetella pertussis*-induced histamine sensitization

Abbas Raza[1], Sean A. Diehl[2], Dimitry N. Krementsov[3], Laure K. Case[4], Dawei Li [5], Jason Kost[6], Robyn L. Ball[4], Elissa J. Chesler [4], Vivek M. Philip[4], Rui Huang[7], Yan Chen [7], Runlin Ma[7], Anna L. Tyler [3], J. Matthew Mahoney [4,8], Elizabeth P. Blankenhorn[9] & Cory Teuscher [1,10 ✉]

Histamine plays pivotal role in normal physiology and dysregulated production of histamine or signaling through histamine receptors (HRH) can promote pathology. Previously, we showed that *Bordetella pertussis* or pertussis toxin can induce histamine sensitization in laboratory inbred mice and is genetically controlled by *Hrh1*/HRH1. HRH1 allotypes differ at three amino acid residues with $P_{263}$-$V_{313}$-$L_{331}$ and $L_{263}$-$M_{313}$-$S_{331}$, imparting sensitization and resistance respectively. Unexpectedly, we found several wild-derived inbred strains that carry the resistant HRH1 allotype ($L_{263}$-$M_{313}$-$S_{331}$) but exhibit histamine sensitization. This suggests the existence of a locus modifying pertussis-dependent histamine sensitization. Congenic mapping identified the location of this modifier locus on mouse chromosome 6 within a functional linkage disequilibrium domain encoding multiple loci controlling sensitization to histamine. We utilized interval-specific single-nucleotide polymorphism (SNP) based association testing across laboratory and wild-derived inbred mouse strains and functional prioritization analyses to identify candidate genes for this modifier locus. *Atg7, Plxnd1, Tmcc1, Mkrn2, Il17re, Pparg, Lhfpl4, Vgll4, Rho* and *Syn2* are candidate genes within this modifier locus, which we named *Bphse*, enhancer of Bordetella pertussis induced histamine sensitization. Taken together, these results identify, using the evolutionarily significant diversity of wild-derived inbred mice, additional genetic mechanisms controlling histamine sensitization.

[1] Department of Medicine, University of Vermont, Burlington, VT 05405, USA. [2] Department of Microbiology and Molecular Genetics, University of Vermont, Burlington, VT 05405, USA. [3] Department of Biomedical and Health Sciences, University of Vermont, Burlington, VT 05405, USA. [4] The Jackson Laboratory, Bar Harbor, ME 04609, USA. [5] Department of Biomedical Science, Florida Atlantic University, Boca Raton, FL 33431, USA. [6] Catalytic Data Science, Charleston, SC 29403, USA. [7] School of Life Sciences, University of the Chinese Academy of Sciences, 100049 Beijing, China. [8] Department of Neurological Sciences, Larner College of Medicine, University of Vermont, Burlington, VT, USA. [9] Department of Microbiology and Immunology, Drexel University College of Medicine, Philadelphia, PA 19129, USA. [10] Pathology and Laboratory Medicine, University of Vermont, Burlington, VT 05405, USA. ✉email: c.teuscher@med.uvm.edu

Histamine (HA) is an endogenous biogenic monoamine important in regulating diverse physiological processes in mice and human. It is synthesized by mast cells, basophils, platelets, neurons, and enterochromaffin-like cells that keep it stored safely within granules[1]. Following cellular activation, HA is released and mediates pleiotropic effects through four designated seven-transmembrane G-protein-coupled receptors (GPCRs): histamine receptor $H_1$-$H_4$ (HRH1, HRH2, HRH3, and HRH4). These are differentially expressed on target cells in various tissues thus influencing a diverse array of physiological processes, including brain function, neurotransmission, secretion of pituitary hormones, cell proliferation and differentiation, hematopoiesis, embryonic development, wound healing and regeneration, and the regulation of gastrointestinal, cardiovascular, and secretory functions[2]. In addition, HA plays a major role in inflammation and the regulation of innate and adaptive immune responses in both normal and pathologic states[3,4].

Historically, HA is most well-known for its role in shock and anaphylaxis[5]. It was first isolated from the parasitic mold ergot of rye (*Claviceps purpurea*) and then synthesized by the decarboxylation of histidine[6–8]. HA occurs naturally in host physiology and can mediate anaphylactic shock, thus understanding regulatory mechanisms of HRH signaling is important[5]. Systemic injection of HA elicits anaphylactic shock-like symptoms in some mammals, including bronchiolar constriction, constricted cardiac and pulmonary arteries, and stimulated cardiac contraction[9–11]. There is significant variability in susceptibility to HA-shock among animal species, with guinea pigs and rabbits being highly susceptible, versus mice and rats which are generally tolerant to in vivo injections of HA[12]. In 1948, Parfentjev *et al.*, found that prior exposure to *Bordetella pertussis* (*B. pertussis*) or purified

pertussis toxin (PTX) overcomes HA resistance among a subset of laboratory derived inbred strains of mice[13]. This phenotype is designated Bphs for *B. pertussis*-induced HA sensitization[14,15]. Bphs-susceptible (Bphs^s) strains die within 30 min following HA injection; which is thought to result from hypotensive and hypovolemic shock while Bphs-resistant (Bphs^r) strains remain healthy[16]. The sensitizing activity elicited by exposure to *B. pertussis* is a function of PTX-catalyzed adenosine diphosphate-ribosylation of the alpha subunit of heterotrimeric guanine nucleotide-binding protein ($G\alpha_{i/o}$), specifically $G\alpha_{i1/3}$[17,18], a guanine nucleotide-binding protein related to the function of the histamine receptor. In addition to HA sensitization, PTX-treated mice exhibit increased systemic vascular permeability and sensitization to other vasoactive amines, such as serotonin (Bpss) and bradykinin (Bpbs)[17,19].

Our previous genetic studies mapped the autosomal dominant *Bphs* locus controlling susceptibility to mouse chromosome 6 (Chr6) and identified it as the structural locus (*Hrh1*) for histamine receptor $H_1$ (HRH1)[15,20]. HRH1 encodes a protein with 488 amino acids (UniProtKB identifiers: P70174; www.uniprot.org). Susceptibility to Bphs segregates with two conserved *Hrh1* haplotypes, mice with the *Hrh1^s* allele (encoding HRH1^s amino acids P, V, P at position 263, 312, and 330 respectively in the primary sequence of HRH1 protein) are sensitive to PTX (Bphs^s) while mice with the *Hrh1^r* allele (encoding HRH1^r amino acids L, M, S at position 263, 312, and 330) are resistant to PTX (Bphs^r). These amino acid changes occur within the third intracellular loop of this G protein-coupled receptor (GPCR): a domain implicated in signal transduction, protein folding, and trafficking. Functionally, both HRH1^s and HRH1^r allelic products (allotypes) equally activate $G\alpha_{q/11}$, the G protein family members that couple HRH1 signaling to second messenger signaling pathways, indicating that the genetic control of susceptibility and resistance to Bphs is not inherently due to differential activation of either $G\alpha_q$ or $G\alpha_{11}$[21]. However, the two HRH1 allotypes exhibit differential cell surface expression and altered intracellular trafficking, with the HRH1^r allele selectively retained within the endoplasmic reticulum (ER). Importantly, all three amino acid residues ($L^{263}$, $M^{312}$, $S^{330}$) comprising the HRH1^r haplotype are required for altered expression[21].

To better understand regulation of loci that could impact Bphs, we carried out an expanded phenotype screen for Bphs susceptibility and resistance that included previously unstudied wild-derived inbred strains of mice, and identified eight Bphs^S strains, despite carrying the Bphs^R *Hrh1^r* allele. Genetic analyses identified a dominant modifying locus linked to *Bphs/Hrh1* capable of complementing *Hrh1^r* to create a Bphs^S phenotype. We have designated this locus *Bphse* for Bphs-enhancer. Interval-specific single-nucleotide polymorphism (SNP) based association testing and functional enrichment is used to identify candidate genes for *Bphse*.

## Results

### *Hrh1* alleles are highly conserved in mice.

We undertook a genetic approach to screen for evolutionarily selected mechanisms that may be capable of modifying the Bphs^r phenotype. Toward this end, we sequenced ~500 bp stretch of genomic DNA encompassing the third intracellular loop of *Hrh1*/HRH1 across 91 laboratory and wild-derived inbred strains of mice (Table 1). Other than the three amino acid changes ($P^{263}$, $V^{312}$, $P^{330} \rightarrow L^{263}$, $M^{312}$, $S^{330}$) described earlier as encoded by Bphs susceptible (*Hrh1^s*) and resistant (*Hrh1^r*) haplotypes[15], we did not identify any additional amino acid changes. Of the 91 strains, 22 carry the *Hrh1^r* allele, whereas 69 carry the *Hrh1^s* allele (Table 1). We next mapped the evolutionary distribution of the two alleles

**Table 1 Distribution of *Hrh1^s* and *Hrh1^r* alleles in laboratory and wild-derived inbred mouse strains.**

| Susceptible haplotype ($P^{263}$, $V^{312}$, $P^{330}$) *Hrh1^s* | | Resistant haplotype ($L^{263}$, $M^{312}$, $S^{330}$) *Hrh1^r* | |
|---|---|---|---|
| 129×1/SvJ | C57BR/cdJ | P/J | AKR/J |
| 129S1/SvImJ | C57L/J | PANCEVO/EiJ | BPL/1J |
| 129T2/SvEmsJ | C58/J | PERA/EiJ | C3H/HeJ |
| A/HeJ | CALB/RkJ | PERC/EiJ | C3H/HeN |
| A/J | CE/E | PL/J | CASA/RkJ |
| A/WySnJ | DBA/1J | RBF/DnJ | CAST/EiJ |
| ALR/LtJ | DBA/2J | RIIIS/J | CBA/J |
| ALS/LtJ | DDY/JclSidSeyFrkJ | SB/LeJ | CBA/N |
| B10.S/DvTee | EL/SuzSeFrkJ | SEA/GnJ | CZECHI/EiJ |
| B10.S/McdgJ | FVB/NCr | SEC/1ReJ | CZECHII/EiJ |
| BALB/cByJ | IS/CamRkJ | SENCARA/PtJ | I/LnJ |
| BALB/cJ | KK/HlJ | SENCARB/PtJ | JF1/Ms |
| BDP/J | LEWES/EiJ | SENCARC/PtJ | MOLC/RkJ |
| BPH/2J | LG/J | SJL/J | MOLD/RkJ |
| BPL/1J | LP/J | SJL/BmJ | MOLF/EiJ |
| BPN/3J | MA/MyJ | SM/J | MRL/MpJ |
| BTBRT+ | MOR/RkJ | SPRET/EiJ | MSM/Ms |
| BXSB/MpJ | NOD/LtJ | ST/BJ | PWD/PhJ |
| C57BL/10J | NON/LtJ | SWR/J | PWK/PhJ |
| C57BL/10SnJ | NOR/LtJ | SWXL-4/TyJ | RF/J |
| C57BL/6ByJ | NZB//BINJ | TIRANO/EiJ | SF/CamEiJ |
| C57BL/6J | NZO/HILtJ | YBR/EiJ | SKIVE/EiJ |
| C57BLKS/J | NZW/LacJ | ZALENDE/EiJ | – |

**Table 2 Bphs susceptibility of mice with the *Hrh1^r* allele.**

| Strain | Group[a] | Histamine (mg/kg) | | | Total | % Aff | *p*-value[b] |
|---|---|---|---|---|---|---|---|
| | | 100 | 50 | 25 | | | |
| C3H/HeJ | 1 | 0/3 | 0/2 | 0/2 | 0/7 | 0 | – |
| C3H/HeN | 1 | 0/2 | 0/2 | 0/2 | 0/6 | 0 | – |
| C3H | – | 0/5 | 0/4 | 0/4 | 0/13 | 0 | – |
| AKR/J | 1 | 1/3 | 0/2 | 0/2 | 1/7 | 14 | 0.4 |
| CAST/EiJ | 7 | 1/3 | 0/3 | 0/3 | 1/9 | 11 | 0.4 |
| CBA/J | 1 | 0/3 | 0/2 | 0/2 | 0/7 | 0 | – |
| CBA/N | 1 | 0/3 | 0/2 | 0/2 | 0/7 | 0 | – |
| CBA | – | 0/6 | 0/4 | 0/4 | 0/14 | 0 | 1.0 |
| I/LnJ | 6 | 2/7 | 0/3 | – | 2/10 | 20 | 0.2 |
| MSM/Ms | 7 | 0/3 | 0/3 | – | 0/6 | 0 | 1.0 |
| MRL/MpJ | 1 | 0/3 | 0/2 | 0/2 | 0/7 | 0 | 1.0 |
| SF/CamEiJ | 1 | 0/4 | 0/2 | – | 0/6 | 0 | 1.0 |
| SKIVE/EiJ | 7 | 2/7 | 1/6 | 0/2 | 3/15 | 20 | 0.2 |
| BPL/1 J | 5 | 1/2 | 2/2 | 2/2 | 5/6 | 83 | 0.0005 |
| CZECHII/EiJ | 7 | 4/4 | 2/4 | 2/2 | 8/10 | 80 | <0.0001 |
| JF1/MsJ | 7 | 2/3 | 2/3 | – | 4/6 | 67 | 0.004 |
| MOLD/EiJ | 7 | 2/2 | 1/2 | 2/2 | 5/6 | 83 | 0.0005 |
| MOLF/EiJ | 7 | 2/2 | 5/5 | 5/5 | 12/12 | 100 | <0.0001 |
| PWD/PhJ | 7 | 5/7 | – | – | 5/7 | 71 | 0.001 |
| PWK/PhJ | 7 | 2/2 | 2/2 | 2/2 | 6/6 | 100 | <0.0001 |
| RF/J | 1 | 2/2 | 2/2 | 2/2 | 6/6 | 86 | <0.0001 |

Mice were injected with 200 ng of PTX on D0. Three days later the animals were challenged with the indicated dose of HA (mg dry weight free base/kg body weight) by intraperitoneal injection and deaths recorded as number of animals dead/number of animals tested at 30 min post HA challenge.
[a]According to mouse family tree (adapted from Petkov et al.[22]).
[b]Relative to C3H mice.

onto a mouse phylogenetic tree (Supplementary Fig. 1)[22]. The *Hrh1^r* allele is primarily restricted to wild-derived group 7 strains and a select sub-branch of group 1 Bagg albino derivatives, whereas the *Hrh1^s* allele is distributed across groups 2–6, which encompasses Swiss mice, Japanese and New Zealand inbred strains, C57/58 strains, Castle mice, C.C. Little DBA and related strains.

Compared to classical inbred strains, wild-derived mice exhibit sequence variation at approximately every 100–200 base pairs and are, in general, more resistant to a variety of pathogens[23–27]. Group 7 strains exemplify this genetic diversity in that it includes representatives of *Mus musculus (M. m) domesticus* (PERA, PERC, WSB, ZALENDE and TIRANO), *M. m. musculus* (PWK, PWD, CZECHI, and CZECHII), *M. m. castaneus* (CAST), *M. m. molossinus* (JF1, MSM, MOLF, MOLD, MOLC), *M. m. hortulanus* (PANCEVO), *M. m. spretus* (SPRET), *M. m. praetextus* (IS), or hybrids of *M. m. musculus* and *M. m. domesticus* (SKIVE), *M. m. musculus* and *M. m. poschiavinus* (RBF) and of *M. m. castaneus* and *M. m. domesticus* (CALB)[28–30]. This genetic variability represents a rich source of evolutionarily selected diversity and has the potential to lead to the identification of genes controlling additional regulatory features arising from host-pathogen co-evolutionary adaptations.

To screen for functional modifying loci capable of complementing *Hrh1^r*, we phenotyped a panel of group 1 (Bagg albino derivatives) and 7 (wild-derived) mice that genotyped as *Hrh1^r*/HRH1^r for susceptibility to Bphs. While nine *Hrh1^r* strains tested were Bphs^r as expected, we found eight that were susceptible to Bphs (Table 2). These Bphs^s strains were confined primarily to group 7 wild-derived strains (Supplementary Fig. 1) in contrast to *Hrh1^r* strains from Group 1 that were mostly Bphs^r. Moreover, there were no segregating structural variants of the *Hrh1* gene (Chr6:114,397,936–114,483,296 bp) between representative Bphs^s and Bphs^r strains of phenotyped Group 1 and Group 7 mice. This is supported by imputed SNP datasets across all phenotyped strains (see methodology) suggesting that the complementing

locus/loci in Group 7 is independent of previously unidentified *Hrh1^s* structural variants.

To confirm the existence of a modifying locus capable of restoring Bphs^s in mice with a *Hrh1^r* allele and to assess its heritability, we selected a subset of Bphs^s-*Hrh1^r* (MOLF, PWK) and Bphs^r-*Hrh1^r* (AKR, CBA, C3H, MRL) strains for follow-up studies. We studied F_1 hybrids between the select strains of interest and *Hrh1*-knockout B6 mice (B6.129P2-*Hrh1^tm1Wtn*/BrenJ, common name H1R KO), which lack a functional *Hrh1^s* gene required for Bphs^s[31]. The H1R KO background, however, could provide potential complementing genetic elements in trans. Both (B6 × H1R KO) F_1 and (C3H.*Bphs*^SJL × H1R KO) F_1 harbor the heterozygous *Hrh1^s/-* allele while (C3H × H1R KO) F_1, (CBA × H1R KO) F_1, (MRL × H1R KO) F_1 and (AKR × H1R KO) F_1 have the *Hrh1^r/-* allele. Both *Hrh1^s*-by- H1R KO F_1 hybrids were Bphs^s, whereas the inbred strain *Hrh1^r*-by- H1R KO F_1 hybrids were Bphs^r (Table 3), in agreement with our prior finding that *Hrh1^s/-* controls susceptibility to Bphs dominantly[14]. In contrast, (MOLF × H1R KO) F_1 and (PWK × H1R KO) F_1 hybrids, that harbor the *Hrh1^r/-* allele, were Bphs^s. This data supports the existence of one or more dominant loci in MOLF and PWK capable of complementing Bphs^r in *Hrh1^r/-* mice.

**A functional linkage disequilibrium domain on Chr6 encodes multiple loci controlling HA-shock.** Given the evidence from inbred strains of mice indicating that a quarter or more of the mammalian genome consists of chromosomal regions containing clusters of functionally related genes, i.e., functional linkage disequilibrium (LD) domains[32,33], we hypothesized that the dominant locus complementing *Hrh1^r* may reside within such a LD domain. Support for the existence of a functional LD domain controlling responsiveness to HA is provided by our recent finding that *Histh1-4*, four QTL on Chr6:45.9-127.9 Mb controlling age- and inflammation-dependent susceptibility to HA-shock in SJL/J, FVB/NJ, NU/J, and SWR/J mice, are in strong LD with *Bphs/Hrh1* (Chr6:114,397,936–114,483,296 bp)[34,35].

**Table 3 Bphs susceptibility of Hrh1ˢ and Hrh1ʳ by H1R KO F₁ hybrid mice.**

| Strain | Histamine (mg/kg) | | | | | Total | % Aff | p-value[a] |
|---|---|---|---|---|---|---|---|---|
| | 100 | 50 | 25 | 12.5 | 6.25 | | | |
| H1R KO | 0/2 | 0/2 | 0/2 | 0/2 | 0/2 | 0/10 | 0 | – |
| C57BL/6J | 3/3 | 3/3 | 3/3 | 2/2 | 1/2 | 12/15 | 90 | <0.0001 |
| (B6 × H1R KO) F₁ | 4/4 | 4/4 | 4/4 | 3/4 | 3/3 | 18/19 | 95 | <0.0001 |
| C3H.Bphs^SJL | 3/3 | 2/2 | 2/2 | 2/2 | 2/2 | 11/11 | 100 | <0.0001 |
| (Bphs^SJL × H1R KO) F₁ | 2/2 | 2/2 | 2/2 | 2/2 | 2/2 | 10/10 | 100 | <0.0001 |
| C3H/HeJ | 1/3 | 0/2 | 0/2 | 0/2 | 0/2 | 1/11 | 9 | – |
| (C3H × H1R KO) F₁ | 0/2 | 0/2 | 0/2 | 0/2 | 0/2 | 0/10 | 0 | – |
| CBA/J | 0/3 | 0/2 | 0/2 | 0/2 | 0/2 | 0/11 | 0 | – |
| (CBA × H1R KO) F₁ | 0/2 | 0/3 | 0/2 | 0/2 | 0/2 | 0/11 | 0 | – |
| AKR/J | 1/3 | 0/2 | 0/2 | 0/2 | 0/2 | 1/11 | 9 | – |
| (AKR × H1R KO) F₁ | 0/2 | 0/2 | 0/2 | 0/2 | 0/2 | 0/10 | 0 | – |
| MRL/MpJ | 0/3 | 0/2 | 0/2 | 0/2 | 0/2 | 0/11 | 0 | – |
| (MRL × H1R KO) F₁ | 2/3 | 0/2 | 0/2 | 0/2 | 0/2 | 2/11 | 18 | – |
| PWK/PhJ | 3/3 | 3/3 | 2/2 | 1/2 | 0/2 | 9/12 | 75 | 0.0005 |
| (PWK × H1R KO) F₁ | 3/3 | 2/2 | 2/2 | 1/2 | 1/2 | 11/13 | 85 | <0.0001 |
| MOLF/MpJ | 2/2 | 2/2 | 2/2 | 2/2 | 0/2 | 8/10 | 80 | <0.0001 |
| (MOLF × H1R KO) F₁ | 2/2 | 2/2 | 2/2 | 2/2 | 0/2 | 8/10 | 80 | <0.0001 |
| (MOLF × H1R KO) × H1R KO | 114 | **Aff** | naff | – | – | – | – | – |
| H1R^−/− | 54 | 0 | 54 | – | – | 0/54 | 0 | – |
| H1R^MOLF/− | 60 | 54 | 6 | – | – | 54/60 | 90 | <0.0001 |

Mice were injected with 200 ng of PTX on D0. Three days later the animals were challenged with the indicated dose of HA (mg dry weight free base/kg body weight) by intraperitoneal injection and deaths recorded as number of animals dead/number of animals tested at 30 min post HA challenge. *Aff* affected animals; *naff* not affected animals.
[a]Relative to HRH1KO mice.

**Table 4 Linkage of Chr6 marker loci to Bphse.**

| Marker | bp | $\chi^2$ | p-value | A Ho | A He | Un Ho | Un He | Total |
|---|---|---|---|---|---|---|---|---|
| rs36385580 | 59,353,905 | 28.8 | 7.95E-08 | 28 | 52 | 65 | 20 | 165 |
| rs38650989 | 72,592,521 | 30.6 | 3.21E-08 | 28 | 52 | 66 | 19 | 165 |
| D6Mit186 | 73,387,511 | 29.5 | 5.49E-08 | 30 | 53 | 66 | 19 | 168 |
| D6Mit102 | 93,463,949 | 38.2 | 6.36E-10 | 25 | 58 | 66 | 19 | 168 |
| D6Mit65 | 101,387,523 | 42.3 | 7.92E-11 | 25 | 58 | 68 | 17 | 168 |
| D6Mit149 | 106,005,405 | 38.5 | 5.44E-10 | 27 | 56 | 68 | 17 | 168 |
| Hrh1 | 114,397,936 | – | – | – | – | – | – | – |
| rs31698248 | 120,207,163 | 41.6 | 1.09E-10 | 26 | 56 | 69 | 16 | 167 |
| D6Mit254 | 125,356,646 | 35.6 | 2.42E-09 | 26 | 56 | 66 | 19 | 167 |
| rs30853093 | 125,365,703 | 34.5 | 4.32E-09 | 26 | 57 | 65 | 20 | 168 |
| rs30662734 | 125,370,997 | 34.5 | 4.32E-09 | 26 | 57 | 65 | 20 | 168 |
| rs36868180 | 127,629,804 | 32.7 | 1.06E-08 | 27 | 56 | 65 | 20 | 168 |
| D6Mit135 | 128,834,894 | 29.2 | 6.53E-08 | 27 | 56 | 63 | 22 | 168 |

Mice were injected with 200 ng of PTX on D0. Three days later the animals were challenged with the indicated dose of HA (mg dry weight free base/kg body weight) by intraperitoneal injection and deaths recorded as number of animals dead/number of animals tested at 30 minutes post HA challenge. Segregation of genotype frequency differences with Bphsˢ (affected = A) and Bphsʳ (unaffected = Un) in (AKR × PWK) × AKR mice were tested by $\chi^2$ in 2 × 2 contingency tables. *He* AKR/PWK allele, *Ho* AKR allele.

To test this, we generated 114 (MOLF × H1R KO) × H1R KO backcross (BC) mice (Table 3), genotyped their *Hrh1* alleles, and challenged them for assessing their Bphs phenotype. As expected, none of the 54 homozygous H1R KO mice phenotyped as Bphsˢ. Of the 114 BC mice studied, 54 (47%) were Bphsˢ, which is consistent with genetic control by a single locus. Furthermore, of the 60 HRH1^MOLF (*Hrh1ʳ*) mice, 54 were Bphsˢ and 6 were Bphsʳ, indicating that the locus capable of complementing Bphsʳ is in fact linked to *Bphs/Hrh1*. We have designated this locus *Bphse* for Bphs-enhancer.

To further test the hypothesis that *Bphse* is linked to *Hrh1*, we generated a cohort of Bphs-phenotyped (AKR × PWK) × AKR BC1 mice and performed linkage analysis using informative markers across a ~70 Mb region encompassing *Hrh1*. Both AKR and PWK mice carry an *Hrh1ʳ* allele; however, unlike inbred AKR mice, which are Bphsʳ, PWK mice are Bphsˢ (Table 2). Overall, 83

of 168 (49%) (AKR × PWK) × AKR BC1 mice were Bphsˢ (Table 4). This phenotype mapped to Chr6: marker loci from rs36385580 thru D6MiT135 (Chr6: 59.3–128.8 Mb) exhibited significant linkage to Bphsˢ with maximal linkage across the 26MB interval bounded by D6MiT102 (Chr6:93,463,949-93,464,093 bp) and rs31698248 (Chr6:120,207,213 bp) which encompasses *Hrh1* (Chr6:114,397,936–114,483,296 bp). This finding is in agreement with the segregation of *Bphse* in (MOLF × H1R KO) × H1R KO BC1 mice and provides further evidence that *Bphse* is linked to *Bphs/Hrh1*.

We next confirmed the existence and physical location of *Bphse* by congenic mapping. Marker-assisted selection was used to introgress the *Bphse^MOLF* and *Bphse^PWK* intervals onto the Bphsʳ C3H and AKR backgrounds, respectively. Starting at N5 through N10, heterozygous and homozygous BC mice were phenotyped for Bphs (Fig. 1). Compared to C3H (C3H and C3H.Bphse^C3H/C3H)

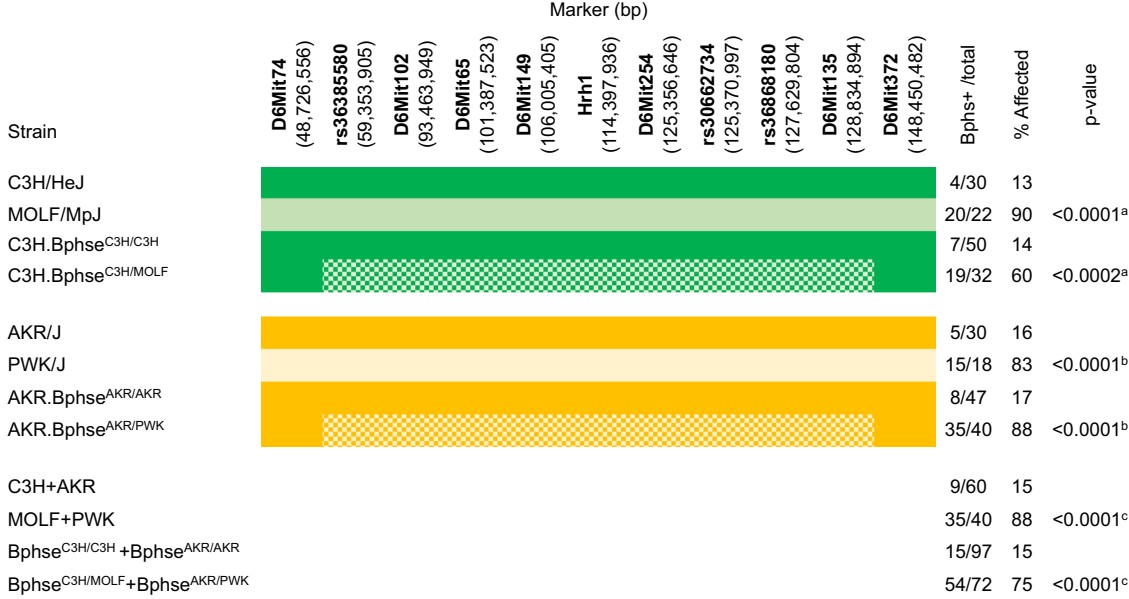

**Fig. 1 Congenic mapping confirms linkage of *Bphse* to *Bphs/Hrh1*.** Mice were injected with 200 ng of PTX on D0. Three days later the animals were challenged with the indicated dose of HA (mg dry weight free base/kg body weight) by intraperitoneal injection and deaths recorded as number of animals dead/number of animals tested at 30 minutes post HA challenge. Marker loci and their location (mm10) are listed along with the respective genotypes: C = C3H, M = MOLF, A = AKR, P = PWK and He = heterozygous. [a]Relative to C3H/HeJ; [b]Relative to AKR/J; [c]Relative to C3H/HeJ and AKR/J combined.

and AKR (AKR and AKR.*Bphse*[AKR/AKR]) mice, both C3H.*Bphse*[C3H/MOLF] and AKR.*Bphse*[AKR/PWK] mice were Bphs[s]. Overall, *Bphse*[C3H/MOLF] and *Bphse*[AKR/PWK] mice were significantly more susceptible to Bphs than *Bphse*[C3H/C3H] and *Bphse*[AKR/AKR] mice ($\chi^2 = 60.63$, df = 1, $p < 0.0001$). The physical mapping results confirm the linkage of *Bphse* to *Bphs/Hrh1* and *Histh1-4* (Chr6:45.9–127.9 Mb)[34,35], and importantly provide strong support for the existence of a functional LD domain on Chr6 encoding multiple loci controlling susceptibility to HA-shock following exposure to environmental factors and infectious agents, including influenza A[35].

**Identification of candidate genes for *Bphse*.** Given that many laboratory and some wild-derived inbred strains have undergone deep sequencing (30–60× genome coverage) with publicly available variant datasets in Mouse Phenome Database (MPD; https://phenome.jax.org/)[36] and Mouse Genomes Project (MGP; https://www.sanger.ac.uk/data/mouse-genomes-project/)[37], we retrieved all coding and non-coding single-nucleotide polymorphism (SNP) data available across the *Bphse* congenic interval (Chr6:59.3–128.8 Mb) among our seventeen Bphs-phenotyped *Hrh1*[r] mouse strains (Table 2). As expected, this dataset lacked SNP coverage for several of the wild-derived inbred strains (>98% SNP missing compared with C57BL/6J) in this study. To complement our dataset, we utilized Chr6 region capture sequencing (see "Methods" section) to sequence and identify SNPs within the *Bphse* interval and integrated these SNPs with the publicly available dataset. This approach yielded a total of 1,303,072 SNPs among which 13,257 SNPs had 100% coverage (no missing genotypes) across all seventeen strains.

To identify variants that segregate with Bphs[s] among *Hrh1*[r] strains, we used efficient mixed-model association (EMMA)[38] and both the larger dataset of 1,303,072 SNPs as well as the smaller dataset of 13,257 SNPs with the anticipation that having complete genotypes would increase the power to detect segregating variants. However, both datasets yielded no significant or suggestive associations which led us to speculate that perhaps the

number of mouse strains available for genetic association analysis may be a limiting factor. To test this, we asked if we could identify any genetic variants that segregate with Bphs[s] independent of *Hrh1* haplotype with the rationale that this approach will identify *Bphs/Hrh1* (positive control) as well as polymorphic gene candidates for *Bphse*. Moreover, *Bphse* expressivity requires the presence of the HRH1 protein, whether encoded by the *Hrh1*[s] or *Hrh1*[r] allele. This method greatly enhanced the number of mouse strains for genetic association analysis, as numerous mouse strains have been phenotyped for Bphs over the years by us and others[15,17,19,39,40].

To accomplish this, we generated SNP datasets as before but across a larger panel of 50 inbred mouse strains (both *Hrh1*[r] and *Hrh1*[s]) (Supplementary Table 1). Using this approach, we identified 3 SNPs in *Atg7* as significant with a stringent cut-off ($p < 3.81 \times 10^{-6}$) and another 163 SNPs in 27 genes with a moderate cut-off ($p < 5.00 \times 10^{-2}$) that were associated with Bphs[s] (Fig. 2a and Supplementary Table 2). There was no difference in predicted candidate genes using either the smaller dataset (13,257 SNPs) or the larger dataset (1,303,072 SNPs). It is important to reiterate that this approach of combining both *Hrh1*[r] and *Hrh1*[s] mouse strains may predict candidates for both *Bphs* and *Bphse*. Thus, *Hrh1*, which is a positive control for this analysis and whose polymorphism has been earlier shown to underlie *Bphs* among laboratory inbred strains[15], was among the candidate genes supporting the predictions from this analysis. We also know that the available SNP data of the entire *Hrh1* gene (~85 kb) among several *Hrh1*[r]/HRH1[r] strains harbor no additional segregating structural variants between Bphs[s] and Bphs[r] mice, excluding involvement of any previously unidentified HRH1 structural alleles underlying *Bphse*.

The *Bphse* predicted candidate genes cluster in a narrow interval of ~5.5 Mb on Chr6:111.0-116.4 Mb (Fig. 2a). Given that our larger dataset (1,303,072 SNPs) includes several thousand SNPs in this shortlisted region of ~5.5 Mb, we asked if we could impute the missing SNPs and do a high-dimensional association run. To impute missing SNPs, we utilized same methodology used in MPD's GenomeMUSter SNP grid[41,42] which in part, uses

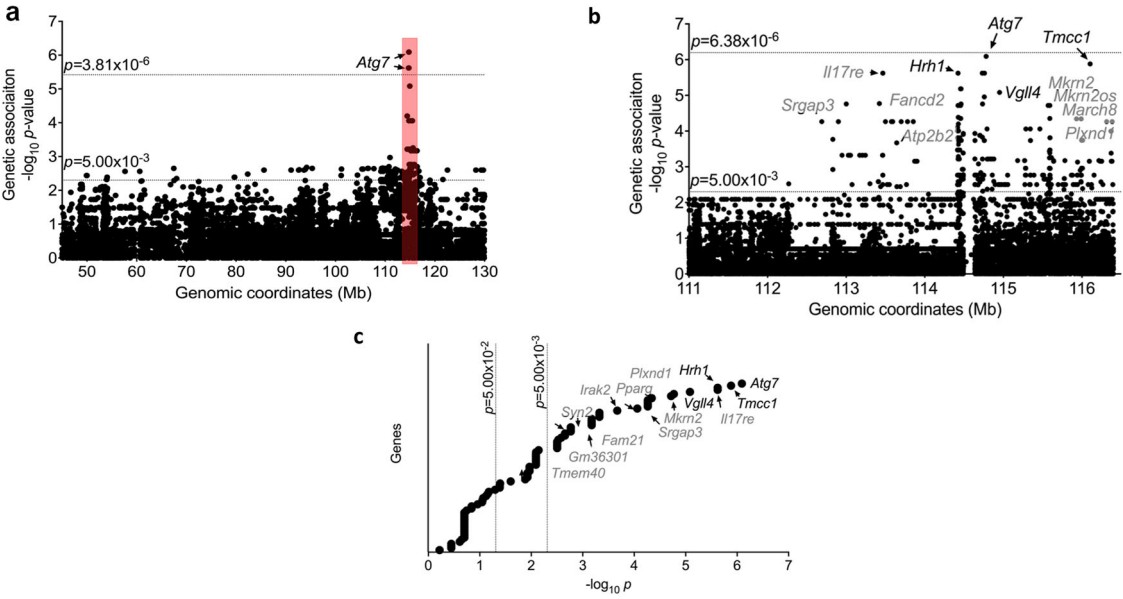

**Fig. 2 Genetic association testing reveals candidates for Bphs.** Bphs[s] was tested using a **a** low-resolution 13,257 SNP panel across Chr6:45.0-130.0 Mb followed by **b** a high-resolution 78,334 imputed SNP panel across Chr6:111.0-116.4 Mb. Both plots show negative log-transformed *p*-values of each SNP tested using Efficient Mixed Model Association (EMMA) on the y-axis. Each filled circle denotes a single SNP. Significance thresholds are shown with a dotted line in each panel. The x-axis denotes Mb coordinates for Chr6 (mm10). **c** Gene names are included for SNPs that crossed *p*-value cut-off of $5.00 \times 10^{-3}$.

the Viterbi algorithm as implemented in HaploQA[43,44] and generated a complete dataset of 78,334 SNPs across Chr6:111.0–116.4 Mb. Using this more complete SNP dataset, we ran a high-resolution genetic association analysis and found *Atg7, Tmcc1, Il17re, Vgll4*, and several others as top hits for *Bphse* ($p < 5.00 \times 10^{-2}$) (Fig. 2b, c and Supplementary Data 1).

As a complementary approach to identify positional candidates for *Bphse*, we employed machine-learning computation, using functional genomic networks[45] to identify network-based signatures of biological association. To this end, we used prior knowledge to generate a list of Bphs-associated biological processes and retrieved gene sets functionally associated with each term. The terms and their justifications are as follows:

- Type I hypersensitivity/anaphylaxis: the death response following systemic HA challenge exhibits symptoms of type I hypersensitivity (T1H)/anaphylaxis including respiratory distress, vasodilation, and anaphylactic shock[46].
- Cardiac: there is evidence suggesting that anaphylactic shock in mice is caused by decreased cardiac output, rather than systemic vasodilation[47].
- Histamine: Bphs is induced by a systemic HA challenge[15].
- G-protein coupled receptor: HRH1 signaling is required for the Bphs phenotype, and all HA receptors belong to the family of G-protein coupled receptors[48].
- Pertussis toxin: Bphs is induced in mouse strains by PTX[12].
- Vascular permeability (VP): hypersensitivity to HA exhibits vascular leakage in skin and muscles[34,35].
- Endoplasmic reticulum (ER)/endoplasmic membrane protein complex (EMC), and endoplasmic reticulum-associated degradation (ERAD): the two HRH1 allotypes exhibit differential protein trafficking and cell surface expression with the HRH1[r] form primarily retained in the ER[21]. The EMC and ERAD are intimately involved in regulating GPCR translocation to the plasma membrane[49,50].

Each of the seven gene sets define a putative Bphs-related process that forms a distinct subnetwork of the full functional genomic network. Using this approach, we identified several hundred genes within the *Bphse* congenic locus that are functionally associated with each biological process, and thus could be gene candidates (Supplementary Data 2).

Genes that are predicted to be highly functionally related to a trait may not have functional variant alleles segregating in the study population and may therefore be unlikely to drive the observed strain differences in Bphs[s]. Using the list of polymorphic genes identified through high-resolution genetic association testing (Fig. 2), we normalized and plotted the respective genetic association score (-$\log_{10} p_{EMMA}$) with functional enrichment ($-\log_{10} FPR$) to focus on genes that overlap both approaches (Fig. 3a). The final ranking was calculated by defining a final gene score ($S_{cg}$, Eq. 1) for each gene, which is the sum of the (normalized) $-\log_{10}(FPR)$ and the $-\log_{10} (p_{EMMA})$ (Fig. 3b). The top ten candidates for *Bphse* as ranked using $S_{cg}$ are: *Atg7, Plxnd1, Tmcc1, Mkrn2, Il17re, Pparg, Lhfpl4, Vgll4, Rho,* and *Syn2*. Furthermore, the predicted candidates for *Bphse* not only overlap *Bphs/Hrh1* but two additional QTLs, *Histh3* (Chr6:99.5–112.3 Mb) and *Histh4* (Chr6:112.3–127.9 Mb), which control age- and inflammation-dependent susceptibility to HA shock in SJL/J, FVB/NJ, NU/J, and SWR/J mice[34,35]. Our findings support the existence of a narrow functional LD domain on Chr6:111.0–116.4 Mb (~5.5 Mb interval) that controls susceptibility to HA-shock elicited by both PTX dependent and independent mechanisms (Supplementary Fig. 2).

## Discussion

*B. pertussis* and PTX elicit a variety of in vivo immunologic and inflammatory responses, including systemic vascular hypersensitivity to serotonin (Bpss), bradykinin (Bpbs), and HA (Bphs)[17]. Utilizing classical laboratory derived inbred strains of mice, we and others showed that susceptibility to Bpss, Bpbs, and Bphs are under unique genetic control with Bpss[s] and Bpbs[s] being recessive traits[17,19] and Bphs controlled by the single autosomal dominant locus *Bphs*[14,51], which we subsequently identified at *Hrh1*[15]. Herein, we present data from several wild-derived inbred

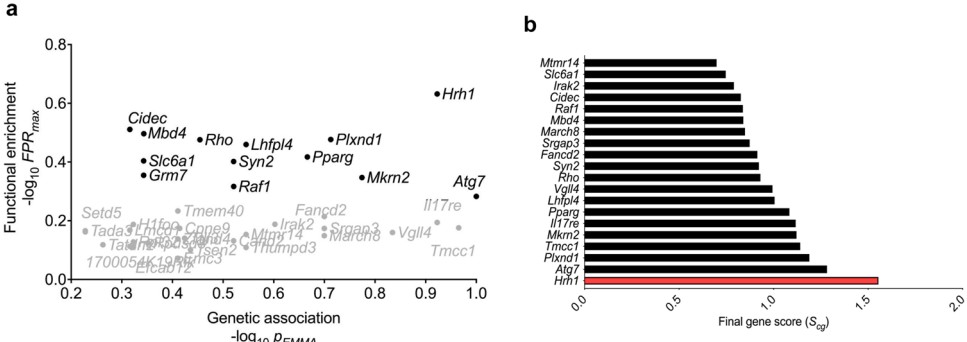

**Fig. 3 Integration of genetic and functional mapping approaches to predict candidates for Bphse. a** The plot shows normalized negative log-transformed false positive rate of maximum functional enrichment (-$\log_{10}$ FPR) on the y-axis. The x-axis denotes the corresponding normalized negative log-transformed genetic association scores. Candidates that were common across both approaches are shown. **b** The list of gene candidates as predicted by both genetic and functional approaches are ranked using a final gene score ($S_{cg}$) on x-axis and the gene names of top 20 candidates listed on y-axis. *Bphs/Hrh1* association is shown in red bar.

strains of mice that harbor the *Hrh1*[r] allele which nevertheless phenotype as Bphs[s]. This is suggestive of the existence of an evolutionarily selected modifying locus that can complement *Hrh1*[r]/HRH1[r] and is supported by the unique phylogenetic distribution of such strains (Supplementary Fig. 1).

Our results with the genetic cross [(MOLF × C3H) × C3H BC mice] confirm the existence of a dominant modifying locus (*Bphse*) capable of complementing Bphs[s] in mice with an *Hrh1*[r] allele. We also found that *Bphse* requires HRH1 protein expression, as no BC1 mice that genotype as *Hrh1*[−/−] were Bphs[s] (Table 3). Among BC1 mice that genotype as *Hrh1*[MOLF/−], only 10% of mice phenotyped as Bphs[r] indicating that *Bphse* is linked with *Bphs/Hrh1*. Linkage scans using microsatellite markers validated significant linkages to Chr6, with maximal significance around *Bphs/Hrh1* (Table 4). In addition, we mapped the physical location of this locus by making congenic mice (C3H.*Bphse*[MOLF+/−] and AKR.*Bphse*[PWK+/−]) that captured the *Bphse* locus on Chr6 (59.3–128.8 Mb, Fig. 1) and replicated the phenotype. To our knowledge, this is the first study assessing Bphs[s] in multiple wild-derived inbred strains of mice, and clearly establish their utility in identifying additional genetic mechanisms controlling HA-shock.

Aside from genetics, several factors could influence HA sensitivity after exposure to *B. pertussis* and PTX including age, sex, and route of sensitization/challenge[12]. In our phenotyping experiments, we used 8–12-week-old mice of each sex and did not find any sex differences. This agrees with earlier studies that found no sex differences in Bphs[s][49]. We also tested the route of administration of PTX and HA challenge using the intraperitoneal and intravenous routes and found no difference. We have not tested the effect of age on Bphs[s] amongst the various strains; however, work from Munoz and others have reported a significant effect of age[12]. It is possible that some of the strains that are Bphs[r] will exhibit Histh[s] as they age or following treatment with complete Freund's adjuvant[34,35].

Our initial mapping of encoding *Bphse* (Chr6:59.3–128.8 Mb) revealed a locus containing hundreds of genes. Until recently, interval specific recombinant congenic mapping was the gold standard to delimit large QTLs associated with a phenotype[50]. Of the thousands of QTLs for various phenotypes and diseases, only a small fraction of genes have been identified through subcongenic mapping, phenotyping and sequencing. The identification of candidate genes from large genomic regions has been revolutionized with the advent of advanced sequencing technologies and genome wide association studies (GWAS)[52]. For example, the Sanger Institute has sequenced 16 inbred laboratory

and wild-derived mouse genomes and The Jackson Laboratory, in conjunction with the University of North Carolina, has genotyped several hundred laboratory inbred strains using the Mouse Diversity Array, that altogether provides an almost complete picture of genetic variation among the various strains. Our approach to identify Bphse candidate genes differs from other mouse GWAS studies to identify disease-associated candidate loci[53]. Instead of running a full genome scan across a large panel of Bphs-phenotyped strains, we tested association of susceptibility exclusively across the *Bphse* locus. This allowed us to use the information gathered from the genetic cross and congenic mapping and delimit the region to be screened for association.

Our first screen using genotype and phenotype data across seventeen *Hrh1*[r] mouse strains did not yield any significant candidates, due to limitation in the number of strains used. To overcome this problem, we excluded *Hrh1* genotype as a covariate. Given that several dozen laboratory inbred mouse strains (Group 2, 3, and 4) have been phenotyped for Bphs[15,17,19,39,40] and also to circumvent the sample size limitation in genetic association testing, we searched for genetic polymorphism across the 50 mouse strains that could explain overall Bphs[s]. *Hrh1*, which is our positive control and associated with Bphs[s] among classical laboratory inbred strains[15], was identified supporting the validity of this approach.

The use of imputed genotypes across 50 phenotyped strains further refined the SNPs associated with Bphs[s]. While every available resource was utilized to generate SNP data across chromosome 6, there remain minute gaps in the coverage e.g. 114438359–114588610 (150 kb) as apparent in Fig. 2 and highlighted in Yang et al., 2011 study[54]. This would preclude any Identity by Descent analysis for this region and is a limitation in this study. As more mouse strains are sequenced in depth, future studies can clarify if there is a unique polymorphism in Chr6: 114438359–114588610 that may explain susceptibility to Bphs.

Recently, a quantitative trait gene prediction tool has been described that utilizes functional genomics information (gene co-expression, protein-protein binding data, ontology annotation and other functional data) to rank candidate genes within large QTLs associated with a respective phenotype[45]. This methodology uses biological prior knowledge to predict candidate genes that could influence multiple pathways affecting the phenotype. We utilized this approach for Bphs, which is known to involve cardiac, vascular, and anaphylactic mechanisms[46,47]. Because the selection of phenotype-associated gene sets is critical for final gene predictions, several terms were used to incorporate sub-phenotypes equivalent to Bphs in the expectation that use of

multiple terms would help identify candidate loci for *Bphse*. Integration of functional predictions with genetic association ($S_{cg}$, Fig. 3) allowed us to focus on only those candidates that reached significance in both approaches.

Given that there is differential cell surface expression of HRH1 depending on the haplotype, it is tempting to speculate that *Bphse* may aid in the folding, trafficking and/or surface delivery of HRH1[53,55]. In this regard, *Tmcc1* (transmembrane and coiled-coil domain family 1) and *Atg7* (autophagy related 7) are promising candidates because of their known roles in protein trafficking in cells. *Tmcc1* is an ER membrane protein that regulates endosome fission and subsequent cargo trafficking to the Golgi[56]. Similarly, *Atg7* is implicated in translocation of cystic fibrosis transmembrane conductance regulator (CFTR) to the surface[57]. Though of different classes, HRH1 and CFTR are multi-pass membrane proteins and may share intracellular trafficking pathways, thus it is tempting to suggest *Atg7* acts in a similar fashion to translocate HRH1 to the surface thereby resulting in the Bphs[s] phenotype. It will be informative to measure the surface expression of HRH1 among the Hrh1[r] strains that phenotype as Bphs[s] (Table 2). Results using bone marrow chimeras suggest that Bphs[s] is a function of the non-hematopoietic compartments[50], so several cell types (endothelial, epithelial, stromal cells) are potential candidates for this cell surface expression analysis.

In addition to *Tmcc1* and *Atg7*, several other predicted candidates for *Bphse* may have potential relevance to phenotypes associated with Bphs including anaphylaxis and mast cell degranulation, and cardiovascular effects (Supplementary Data 2). Proliferator-activated receptor-gamma (*Pparg*) encodes a nuclear receptor protein belonging to the peroxisome proliferator-activated receptor (Ppar) family. Activation of PPARγ suppresses mast cell maturation and is involved in allergic disease[58,59]. Because mast cells are major drivers of pathological events in anaphylaxis[58], *Pparg* may be highly relevant to Bphs. In addition, increased PPARγ expression is associated with cardiac dysfunction[60].

Given that this study has shortlisted candidates for Bphse to 10 candidates, it is important to highlight that this is not an exhaustive list of potential modifiers (enhancers, non-coding RNAs, genes with unknown function, etc.) and follow-up studies are needed to determine the causal loci for Bphse. One way to test this would be to quantify the mRNA expression of some of these top candidates between Bphs[s] and Bphs[r] strains and investigate whether they interact with HRH1.

Importantly, the fact that *Bphse* resides within a smaller functional LD that includes *Histh3* and *Histh4* (Supplementary Fig 2) is of potential clinical significance. *Histh* is an autosomal recessive genetic locus that controls susceptibility to *B. pertussis* and PTX-independent, age- and inflammation-dependent HA-shock in SJL/J mice[34,35]. Four sub-QTLs (*Histh1-4*) define *Histh*, each contributing 17%, 19%, 14%, and 10%, respectively, to the overall penetrance of Histh. Importantly, *Histh* is syntenic to the genomic locus most strongly associated with systemic capillary leak syndrome (SCLS) in humans (3p25.3). SCLS or Clarkson disease is a rare disease of unknown etiology characterized by recurrent episodes of vascular leakage of proteins and fluids into peripheral tissues, resulting in whole-body edema and hypotensive shock. Additionally, Histh[s] SJL/J mice recapitulate many of the cardinal features of SCLS, including susceptibility to HA- and infection-triggered vascular leak and the clinical diagnostic triad of hypotension, elevated hematocrit, and hypoalbuminemia and as such makes them a natural occurring animal model for SCLS[34,35]. Clearly, detailed genetic analysis and identification of the causative genes underlying *Bphse*, *Histh3*, and *Histh4* may reveal orthologous candidate genes and or pathways that contribute not only to SCLS, but also to normal and dysregulated mechanisms underlying vascular barrier function more generally.

## Methods

**Animals.** A large number of diverse mice strains from *Mus musculus (M. m) domesticus* (WSB), *M. m. musculus* (PWK, PWD, CZECHI, and CZECHII), *M. m. castaneus* (CAST), *M. m. molossinus* (JF1, MSM, MOLF, MOLD, MOLC), *M. m. hortulanus* (PANCEVO), *M. m. spretus* (SPRET), *M. m. praetextus* (IS), or hybrids of *M. m. musculus* and *M. m. domesticus* (SKIVE), *M. m. musculus* and *M. m. poschiavinus* (RBF) and of *M. m. castaneus* and *M. m. domesticus* (CALB) as well as AKR/J (AKR), BPL/1J, C3H/HeJ (C3H), C3H/HeN, CAST/EiJ, C57BL/6J (B6), CBA/J (CBA), CBA/N, CZECHII/EiJ, I/LnJ, JF1/MsJ, MOLD/EiJ, MOLF/EiJ (MOLF), MRL/MpJ (MRL), MSM/Ms, PWD/PhJ, PWK/PhJ (PWK), RF/J, SF/CamEiJ, and SKIVE/EiJ were purchased from the Jackson Laboratory (Bar Harbor, Maine). The age of animals was between 8–14 weeks and were rested for 2 weeks prior to any experiments. Where possible the minimum number of animals per genotype was kept at 3 animals per sex and are listed in each table. B6.129P-*Hrh1*[tm1Wat] (H1R KO)[31], C3H.*Bphs*[SJL] (C3H.*Bphs*[S])[15], (B6 × H1R KO) F[1], (C3H × H1R KO) F[1], (CBA × H1R KO) F[1], (AKR × H1R KO) F[1], (MRL × H1R KO) F[1], (AKR × PWK) F[1], (C3H × MOLF) F[1], (MOLF × H1R KO) × H1R KO, (AKR × PWK) × AKR, (C3H × MOLF) × C3H, C3H.*Bphs*[MOLF+/−], C3H.*Bphse*[C3H], AKR.*Bphse*[PWK+/−] and AKR.*Bphse*[AKR] were generated and maintained under specific pathogen free conditions in the vivarium of the Given Medical Building at the University of Vermont according to National Institutes of Health guidelines. All animal studies were approved by the Institutional Animal Care and Use Committee of the University of Vermont.

**Bphs phenotyping.** Bphs phenotyping was carried out as previously described[15]. Briefly, mice were injected with purified PTX (List Biological Laboratories, Inc.) in 0.025 M Tris buffer containing 0.5 M NaCl and 0.017% Triton X-100, pH 7.6. Control animals received carrier alone. Three days later, mice were challenged by injection with histamine (milligrams per kilogram of body weight [dry weight], free base) suspended in phosphate-buffered saline (PBS). Deaths were recorded at 30 min post-challenge. The results are expressed as the number of animals dead over the number of animals studied.

**DNA sequencing of the third intracellular loop of *Hrh1*.** DNA for 91 inbred laboratory and wild-derived strains of mice was purchased from the Mouse DNA resource at The Jackson Laboratory (www.jax.org) and used in an *Hrh1* specific PCR reaction using the following primer sets: forward-740F, 5′-TGCCAA-GAAACCTGGGAAAG-3′, and reverse-1250R, 5′-CAACTGCTTGGCTGCCTTC-3′ that amplify the third intracellular loop of *Hrh1*. Thermocycling was carried out for a 15 µl reaction mix with 2 mM MgCl2, 200 µM dNTPs, 0.2 µM primers, 1 unit of Taq polymerase, and ~50 ng of genomic DNA together with an initial 2-min 97 °C denaturation followed by 35 cycles of 97 °C for 30 s, 58 °C for 30 s, and 72 °C for 30 s. The final extension was for 5 min at 72 °C. *Hrh1* amplicons from each mouse strain were gel purified (Qiagen Cat# 28115) and DNA sequencing reactions were performed with the BigDye terminator cycle sequencing kit (Applied Biosystems, Foster City, CA) using 740 F or 1250 F reverse primers. The reaction products were resolved on an ABI Prism 3100 DNA sequencer at the DNA analysis facility at the University of Vermont. DNA sequencing data were assembled and analyzed using MultiAlign[61]. Each potential nucleotide sequence polymorphism was confirmed with chromatographic sequencing profiles using Chromas v2.6.5 (https://technelysium.com.au/wp/)

**DNA isolation and genotyping.** DNA was isolated from mouse tail clippings as previously described[13]. Briefly, individual tail clippings were incubated with cell lysis buffer (125 mg/ml proteinase K, 100 mM NaCl, 10 mM Tris-HCl (pH 8.3), 10 mM EDTA, 100 mM KCl, 0.50% SDS, 300 ml) overnight at 55 °C. The next day, 6 M NaCl (150 ml) was added followed by centrifugation for 10 min at 4 °C. The supernatant layer was transferred to a fresh tube containing 300 µl isopropanol. After centrifuging for 2 min, the supernatant was discarded, and the pellet washed with 70% ethanol. After a final 2 min centrifugation, the supernatant was discarded, and DNA was air dried and resuspended in TE. Genotyping was performed using microsatellite, sequence specific, and *Hrh1* primers (Supplementary Table 3).

**Microsatellite primers.** Polymorphic microsatellites were selected to have a minimum polymorphism of 8 bp for optimal identification by agarose gel electrophoresis. Briefly, primers were synthesized by IDT-DNA (Coralville, IA) and diluted to a concentration of 10 µM. PCR amplification was performed using Promega GoTaq according to standard conditions and amplicons were subjected to 2% agarose gel electrophoresis and visualized by ethidium bromide and UV light.

**Sequence-specific primers.** Genotyping was performed using sequence specific primers that differ only at the 3′ nucleotide corresponding to each allele of the identified SNP[62]. Each primer set was designed using Primer3 to have a Tmelt of 58–60 °C and synthesized by IDT-DNA (Coralville, IA) and used at a concentration of 100 µM. PCR reactions were subjected to cycling conditions as described and if found to be necessary, the annealing temperature at each stage was adjusted to accommodate the optimal Tmelt. Amplicons were electrophoresed with 10 µl Orange G loading buffer on a 1.5% agarose gel stained with ethidium bromide and visualized by UV light. The presence of a SNP specific allele was scored by observing an amplicon of the expected size in either reaction.

**HRH1KO mice genotyping.** Wild-type and $Hrh1^{-/-}$ alleles were genotyped as previously described (Supplementary Table 3)[15]. Approximately 60 ng of DNA was amplified (GeneAmp PCR system 9700, Applied Biosystems, Foster City, CA). The DNA was amplified by incubation at 94 °C for 3 min followed by 35 cycles of 94 °C for 30 s, 62 °C for 30 s, and 72 °C for 30 s. At the end of the 35 cycles, the DNA was incubated at 72 °C for 10 min and 4 °C for 10 min. The amplified DNA was analyzed by gel electrophoresis in a 1.5% agarose gel. The DNA was visualized by staining with ethidium bromide.

**Linkage analysis and generation of *Bphse* congenic.** Segregation of genotype frequency differences with susceptibility and resistance to Bphs in (MOLF × H1R KO) × H1R KO and (AKR × PWK) × AKR mice were tested by χ² in 2 × 2 contingency tables. C3H.*Bphse*^MOLF+/-, C3H.*Bphse*^C3H, AKR.*Bphse*^PWK+/- and AKR.*Bphse*^AKR congenic mice were derived by marker assisted selection. (AKR × PWK) × AKR and (C3H × MOLF) × C3H mice that were heterozygous across the *Bphse* interval at N2 and at each successive BC generation were selected for continued breeding. *Bphse* congenic mice were maintained as heterozygotes.

**Low-resolution interval-specific targeted genetic association testing.** Genotype data (SNPs in both coding and non-coding) of 50 mouse strains (Supplementary Table 1) that were phenotyped for Bphs either by us or described in the literature[12,15,17,19,40], was retrieved from public databases at the Sanger Institute (https://www.sanger.ac.uk/science/data/mouse-genomes-project) and The Jackson Laboratory (https://phenome.jax.org/). The lack of representation of several inbred strains especially wild-derived strains in these databases were compensated by genotyping using chromosome region capture sequencing[62]. *DNA Fragmentation*: For chromosome region capture sequencing, 3 μg of genomic DNA from BPN/3J, BPL/1J, CASA/RkJ, CAST/EiJ, CBA/J, C3H/HeN, CZECHII/EiJ, JF1/Ms, MOLD/EiJ, MOLF/EiJ, MRL/MpJ, MSM/Ms, NU/J, PWD/PhJ, SF/CamEiJ, and SKIVE/EiJ mice was sheared into fragments of ~200 bp with the Covaris E220 system (Covaris, USA). The sheared DNA fragments were then purified for each of the 16 mice strains using AMPure XP Beads (Beckman, USA), following the instructions of the reagent supplier. DNA Library Construction: DNA libraries of the purified fragments were constructed with SureSelect Library Prep Kit (Agilent, USA). In brief, DNA end-repair was performed for the fragments from each mouse strain using 1x End Repair Buffer (NEBNext® End Repair Reaction Buffer New England Biolabs), dNTP Mix, T4 DNA Polymerase, Klenow DNA Polymerase, and T4 Polynucleotide Kinase. After incubation of the mixture at 20 °C for 30 min, addition of nucleotide A at the 3′ end of the sequence was performed using 10×Klenow Polymerase Buffer, dATP and Exo(-) Klenow, at 37 °C for 30 min. Ligation reaction was then conducted using T4 DNA Ligase Buffer, SureSelect Adaptor Oligo Mix, and T4 DNA Ligase at 20 °C for 15 min. The adaptor-ligated library was amplified using SureSelect Primer, SureSelect ILM Indexing Pre Capture PCR Reverse Primer, 5X Herculase II Rxn Buffer, 100 mM dNTP Mix, and Herculase II Fusion DNA Polymerase. Amplification conditions were: initial denaturation at 98 °C for 2 min then 30 cycles of 98 °C for 30 s, 65 °C annealing for 30 s, and 72 °C extension for 30 s. Purification of the amplified products was performed with 1.8X Agencourt AMPure XP beads for each of the libraries. The average insert length for the adaptor-ligated libraries ranged between 225–275 bp. Hybridization capture: The libraries were subjected for the hybridization capture using the SureSelect Target Enrichment Kit (Agilent, USA), following the instruction of the reagent supplier. The prepared library processed with SureSelect Block Mix at 95 °C for 5 min, followed by holding at 65 °C, and the Hybridization Buffer plus capture library mix were added and maintained at 65 °C for 24 h. Finally, Dynabeads M-280 streptavidin (Life, USA) was used for the enrichment of the Captured DNA libraries[63,64]. Index amplification: for each enriched captured DNA library, the index amplification was performed with 5X Herculase II Rxn Buffer, 100 mM dNTP Mix, SureSelect ILM Indexing Post Capture Forward PCR Primer, and Herculase II Fusion DNA Polymerase. The reaction procedure was: 98 °C Pre-denaturation for 2 min, 98 °C denaturation for 30 s, 57 °C annealing for 30 s, 72 °C extension for 30 s, amplification for 12 rounds, followed by purification using 1.8 times the volume of AMPure XP Beads. DNA libraries of 250-350 bp range were obtained for the subsequent sequencing[63]. DNA Sequencing: a 10 ng library was used for cluster generation in cBot with the TruSeq PE Cluster Kit (Illumina, USA) followed by bidirectional sequencing in an Illumina Hiseq 2500 to obtain the data of 2 × 150 bp. Data processing and SNP calls: to ensure the quality of subsequent information analysis, the original sequence was filtered with the software SolexaQA to get high-quality Clean Reads[65]. Efficient high-quality sequencing data was mapped to the reference genome mm10 by the BWA software[66], samtools[67] was used for sorting, picard tools was used for duplicate read removal, and GATK was used for realignment around indels and base quality score recalibration[54]. Finally, GATK HaplotypeCaller was used for SNP detection.

All SNP datasets (MPD, Chromosome region capture sequencing) were collated to yield a total of 1,303,072 SNPs across the Bphse interval (Chr6: 59–129 Mb), among which 13,257 SNPs had 100% coverage (no missing genotypes) across all strains.

To calculate associations between genetic polymorphisms and Bphs, we used efficient mixed-model association (EMMA)[38]. This method treats genetic relatedness as a random variable in a linear mixed model to account for population structure, thereby reducing false associations between SNPs and the measured trait.

We used the likelihood ratio test function (emma.ML.LRT) to generate *p*-values. Significance was assessed with Bonferroni multiple correction testing. The -log transformed *p*-values were plotted using GraphPad Prism7 and genomic coordinates included for each SNP using the latest mouse genome build GRCm38.p5/mm10.

**Genotype imputation methodology.** To impute genotypes, we used the same methodology employed by MPD's GenomeMUSter SNP grid[41,42]. A merged SNP dataset over the Chr6 region 111.0-116.4 Mb (GRCm38/mm10 and dbSNP build 142) was constructed from the 11 SNP genotyping datasets available (see Supplementary Table 4) on the Mouse Phenome Database (MPD)[43,44,68–73]. This MPD-derived merged genotype dataset of 577 strains was then merged with the genotype data for 50 strains generated earlier for low-resolution mapping (see Methods). We leveraged the genotype data from over 577 strains to impute genotypes for missing SNP states across the region for the 50 strains of interest (see Supplementary Table 1). To impute genotypes on a target strain, we utilized the Viterbi algorithm implemented in HaploQA[43,44] where the input was a subset of strains most phylogenetically similar to the target strain. This imputation strategy resulted in a dataset of 78,334 SNPs in the genomic region of interest. Across all 50 strains, the median number of missing SNP genotypes after imputation was 2.45% with a maximum missing of only 5.4% for one of the strains.

**Trait-related gene sets.** The positional candidate genes were ranked based on their predicted association with seven functional terms related to the Bphs phenotype: Cardiac, G-protein coupled receptor, Histamine, Pertussis toxin, Type I hypersensitivity, Vascular Permeability, and ER/EMC/ERAD. Gene Weaver[74] was used to identify genes annotated with each term. Each term was entered the Gene Weaver homepage (https://geneweaver.org) and search restricted to human, rat, and mouse genes, and to curated lists only. Mouse homologs for each gene were retrieved using the batch query tool in MGI (http://www.informatics.jax.org/batch_data.shtml). In addition, using the Gene Expression Omnibus (GEO) and PubMed, additional gene expression data sets were retrieved for each phenotype term. Final gene lists consisted of the unique set of genes associated with each process term.

**Functional enrichment and ranking of Bphs associated genes.** We associated genes with Bphs-related functions as described in Tyler et al.[34]. Briefly, we used the connectivity weights in the Functional Network of Tissues in Mouse (FNTM) as features for training support vector machines[75]. Each feature consisted of the connection weights from a given gene to genes in the functional module. To improve classification and reduce over-generalization we clustered each functional gene set into modules, each <400 genes[45]. For each of these modules, we trained 100 SVMs to classify the module genes from a balanced set of randomly chosen genes from outside the module. We used 10-fold cross validation and a linear kernel. We also trained each SVM over a series of cost parameters identified by iteratively narrowing the cost parameter window to identify a series of eight cost parameters that maximized classification accuracy. We then used the training modules to score each positional candidate gene in the *Bphse* locus. To compare scores across multiple trained models, we converted SVM scores to false positive rates.

**Combined gene score.** To create the final ranked list of positional candidate genes, we combined the SNP association scores with the functional predictions derived from the SVMs. We scaled each of these scores by its maximum value across all positional candidates and summed them together to derive a combined gene score ($S_{cg}$) that incorporated both functional predictions and genetic influence:

$$S_{cg} = \frac{-\log_{10}(\rho EMMA)}{\max\limits_{pos.cand.} - \log_{10}(\rho EMMA)} + \frac{-\log_{10}(FPR_{SVM})}{\max\limits_{pos.cand.} - \log_{10}(FPR_{SVM})} \quad (1)$$

where the denominators of the two terms on the right-hand side are the maximum values of -$\log_{10}(pEMMA)$ and -$\log_{10}(FPR_{SVM})$ over all positional candidates in *Bphse*, respectively, which normalizes the functional and positional scores to each other. SNPs were assigned to the nearest gene within 1 Mb. If more than one SNP was assigned to a gene, we used the maximum negative $\log_{10} p$ value among all SNPs assigned to the gene.

**Statistics and reproducibility.** Unless otherwise noted, experiments were repeated at least 3 times, and data are presented as aggregate sum of all experiments. Data were analyzed using *t* test or ANOVA by PRISM (GraphPad) as indicated in the figure legends. *p* values < 0.05 were considered statistically significant.

**Reporting summary.** Further information on research design is available in the Nature Portfolio Reporting Summary linked to this article.

## Data availability

All data generated or analysed during this study are included in this published article, Supplementary Information, and Supplementary Data. The Source data underlying the

plots shown in Figs. 2–3 are provided in Supplementary Data 2–3. This trait can be accessed under the name Histh. SNP data were retrieved from publicly available databases (mouse phenome database, https://phenome.jax.org/snp/retrievals; Mouse genomes project, https://www.sanger.ac.uk/sanger/Mouse_SnpViewer/rel-1505). A list of studies that have been used to generate phenotypic data are available at: https://phenome.jax.org/about/snp_retrievals_help.

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

## Acknowledgements

Anna Tyler and J. Matthew Mahoney were supported by a grant (R21 LM012615; ALT and JMM) from the National Library of Medicine and a grant (P20GM130454; JMM) from the National Institute of General Medical Sciences of the United States National Institutes of Health (NIH). Abbas Raza, Dimitry Krementsov, Elizabeth Blankenhorn, and Cory Teuscher were supported by grants from the NIH and the National Multiple Sclerosis Society (NMSS). Elissa Chesler, Vivek Philip and the Mouse Phenome Database are supported by R01 DA028420. Robyn Ball and Elissa Chesler are supported by U54 OD030187. Dimitry Krementsov was supported by NIH grants from the National Institute of Neurological Disease and Stroke (R01 NS097596), National Institute of Allergy and Infectious Disease (R21 AI145306), and the NMSS (RR-1602-07780).

We would like to thank Keith Sheppard and Molly Bogue for their assistance with genomic imputation as well as the Mouse Phenome Database web resource (RRID:SCR_003212) and the JAX Cancer Center Support Grant. We also are grateful for the assistance of Laura Cort in the genotyping of the backcross progeny.

## Author contributions

A.R., D.N.K., L.K.C., and C.T. performed experiments, analyzed data, and wrote the paper. S.A.D., R.H., Y.C., R.M., A.L.T, J.M.M., and E.P.B. performed experiments and analyzed data. D.L., J.K., R.L.B., E.J.C., and V.M.P. analyzed data.

## Competing interests

The authors declare no competing interests.
