## [Peer Review File · Communications Biology]

Reviewers' comments:

Reviewer #1 (Remarks to the Author):

Brief summary of the manuscript

In this investigation on histamine sensitization, it was found that mice with the histamine sensitization resistant alleles were still susceptible to histamine shock. A targeted genetic mapping approach using 50 strains of mice revealed a locus on chromosome 6 that is associated with histamine sensitization. This modifier locus on chr 6 likely acts in an epistatic interaction with the histamine receptor. Interestingly, *Hrh1* and this new Bphs-enhancer region are very close together. Candidate genes within this locus were identified, paving the way for additional studies on histamine shock.

Overall impression of the work

This is an important study showing the genetic mechanisms of histamine shock are more complex than previously believed. The identification of this secondary locus (and maybe more) that can have a dominant effect making a strain carrying the resistant allele susceptible is novel. Candidate genes are identified, setting up future studies in this area.

I was at first concerned that this was a genetic mapping study using 50 strains that did not take into account the genetic relatedness between the related strains, but I was glad to see the authors use a tool like EMMA to account for population structure. However, chromosome 6 has several IBD regions that can make fine scale genetic mapping difficult. This is due to the fact that many of the classical mouse strains used in this study have limited genetic diversity. The wild derived strains add to the genetic diversity, but are lacking in other areas like available sequence data. Future experiments using CC, DO, or BXD mouse panels could be more powerful, but the current design seems to still identify a locus of interest.

I do wonder if the Bphse region is just picking up on a non-coding variant in a promoter or enhancer of *Hrh1*, or could be in a splice site mutation that reduces *Hrh1* expression. I think if the discussion was expanded to include this possibility, it would improve the manuscript.

Specific comments, with recommendations for addressing each comment

1 Page 7. The discussion on the evolutionary distribution of the two alleles using the mouse phylogenetic tree from Petkov et al 2004 may not be the most appropriate illustration. This is because the classical inbred strains have limited and non-randomly distributed genetic diversity, and the segregation of a particular locus may not follow the pattern of global genomic relatedness. Please review Yang et al. 2011 (<https://dx.doi.org/10.1534/genetics.106.067868>) as they illustrate the non randomness of the mouse chromosome 6 in Figure 2. (I will attach this figure as well)

From the sequence analysis performed later, is it possible to reconstruct the relatedness between strains by origin or as haplotypes (i.e. m. mus, m. cast, m. dom) at this Bphse locus on chromosome 6, as done in the attached figure, and then highlight the bphs susceptibility? If not, that is ok, but the mouse family tree in supplemental figure 1 does not represent the genomic relatedness of chromosome 6.

2 I see in figure 2B a band with 0 genetic variation between 114 and 115 that is likely one region with high identical-by-descent found in Yang et al. 2011. Interesting that *Hrh1* sits on just the edge of this region. **Could there be strain specific variation in this IBD region that was missed when you used the imputed genotypes? This may be a concern as lab mouse strains are maintained with small breeding populations and can easily acquire and maintain novel mutations.**

3 Page 12. You say that you know all the segregating nonsynonymous structural variants, **but could there be variants in promoter and enhancer regions affecting expression? How do you know that the Bphse locus, which contains *Hrh1*, isn't just picking up on non-coding variation?**

4 **Have you looked at expression levels of *Hrh1* across available strains and compared it to**

your recorded resistant and sensitive phenotypes? Yu et al. 2020 and Gu et al. 2020 (doi: 10.1111/epi.16741, doi: 10.1111/epi.16617) recently showed that a splice site variant in Gabra2 (found in C57BL/6J!) reduced expression of this gene, leading to increased sensitivity to epilepsy. I really wonder if your SNP analysis is picking up on something like this. Can you rule this out in your study by looking at gene expression? I think you at least need to bring up this point in your discussion.

5 Page 22. "Control animals received carrier." should be "A carrier"?

6 Page 22 "DNA sequencing of third intracellular loop of Hrh1." should be of THE third?

7 Page 24 " using Promega GoTaq according standard conditions " should be corrected.

8 Page 26 "The prepared library were then reacted with.."should be corrected.

9 Page 30. Maybe this is an error with the pdf, but I don't see a formula used for the combined genetic score.

10 Figure 2B, is it possible to display the gene names in a more organized way? Maybe it doesn't matter so much if the same ones are represented in Fig 2C. If that is the case, just remove the names in 2B.

11 Again, similar to Figure 3A. It is difficult to read the gene names here. They either need a smaller font, or should be organized differently because it is not easy to read.

12 Supplemental Figure 2 is impossible to read. Could this just be presented in a table?

Reviewer #2 (Remarks to the Author):

The manuscript by Raza et al. reports a robust genetic analysis that uncovers a novel element regulating histamine sensitization following Bordetella pertussis infection and and pertussis toxin action.

The authors present a state-of-the-art mapping and analysis of a quantitative trait locus that modulates sensitivity to pertussis toxin action-induced histamine sensitization. Combining classical backcross and congenic mapping with functional linkage disequilibrium analysis, the authors identified a 5.5 Mb interval that may carry a dominant regulator of Bordetella pertussis histamine sensitization by the Hrh1 gene (Bphs-se). Ten most probable candidate genes were identified within the interval. The manuscript is importantly extending our knowledge on genetic control of sensitivity to histamine. A particular strength of this work lies in the combination of computational approaches with analysis of diversity of wild-derived inbred mice. However, presentation of the results and genetic terminology should be improved and some questions are to be answered in order to fully serve the community.

Here are my points:

Abstract and other places – there is a terminology mix-up. Bphs is defined as a phenotype and not as a gene. Hence, using terms like Bphss/Bphsr alleles represents an oxymoron that needs to be sorted out.

p. 5 - and other places. Terminology Hrh1r/HRH1r allele is used, where HRH1 is the gene product, protein, not the allele of Hrh1 gene. Correct and improve terminology, such shortcutting is unacceptable.

p.7 - The authors sequenced 500 bp stretches of genomic DNA, whereas the HRH1 protein has 488 residues. Can the authors exclude that some Bphss strains genotyped as Hrh1r are in fact Hrh1s due to undiscovered polymorphisms outside of the sequenced 500 nt interval? Please, explain, elaborate on that in Result and in Discussion.

P.8 - comparison of the entire Hrh1 gene (Chr6:114,397,936-114,483,296 bp) between several of the Group 1 and Group 7 phenotyped strains found no segregating non-synonymous structural variants (data not shown). Does that mean that all phenotypically Bphs strains that were

genotyped as Hrh1r, were included in sequencing of the entire genes? This should be specified.
p. 8. HRH1KO is a non-standard designation of the Hrh1 knockout B6 strain. Its proper name should be at least once introduced to follow the rules of the International Committee on Standardized Genetic Nomenclature for Mice. It would also help to show a scheme with all the used combinations of Hrh1 and Bphs-e genotypes of the strains used in the backcrosses.

P.8 Is there any phenotype difference between Bphs sensitivity of HRH1KO/KO and Hrh1s/ Hrh1s?

P.9. Six out of 60 of offspring of (MOLF × HRH1KO) × HRH1KO backcross were apparent recombinants between the Hrh1 gene and Bphs-e locus suggesting 10cM or cca 20 Mb distance between them. But such conclusion would contradict the colocalization of Hrh1 and Bphs within a 5 Mb interval. Please explain/discuss.

The authors need to acknowledge that the found 10 candidates in the 5 Mb segment is not an exhaustive list of potential modifiers (enhancers, non-coding RNAs, gene models with unknown function, etc.)

Reviewer #3 (Remarks to the Author):

This paper by Raza et al defines the identification histamine sensitization mechanisms in the mouse population for inbred and outbred wild animals. The data within this paper are exciting findings and deserving of publication in a good journal. The writing however is not. The logic framing in the story sells short the discoveries and importance. The paper is very tough to read. The abstract is too vague while not clearly establishing the focus of the work. The use of abbreviations so similar make this a very tough read in addition to the overuse of mouse strains in sentences. The introduction is tough to follow. The results have too many hand waving statements about why something was done. When a first sentence needs a reference to it, it detracts from presenting the results. Many of those cited statements I also disagree with them. The writing needs to be more concise and focused on just presenting the outstanding data in as simple of way as possible. Avoid the continued jargon on the mouse field if this is targeted for the broad audience of this journal. This is a simple study with outstanding data. I have no issues with any experiments or data, just how they are presented. Without substantial rewriting, this is a paper that should only be published in a journal dedicated to mouse researchers.

Abstract: In general the abstract uses too many vague descriptions and is very diffuse to what the actual results are, making it a bit frustrating to read, such that one needs to go to the full paper to have any idea what is actually being done. As the work is strong, the abstract needs a nearly complete rewrite. Need to define the subset of mice used to show susceptibility. What is the exact allele driving change? The "extended panel" needs to be defined. Exactly how many of them. Are these a traditional panel? The "eight strains" is vague, are these 8 related strains and thus could be referred to instead based on the evolutionary divergence points? "Genetic analyses" is too vague. What does that mean. Narrowing to a 5.5 Mb region does not seem publishable as a major discovery to end the abstract on. The end of the abstract leaves more work than what the start of the abstract set out to do. This loses me and detracts from the awesome work in the paper itself. It makes more sense to frame this paper as setting out to discover if other alleles outside those known can impact histamine sensitization, where the conclusion is a resounding yes. That does not leave the story hanging for so much follow up validation.

Introduction: The introduction is filled with poor writing and grammatical issues. It is very tough to read. This is so focused on the mouse that it is very unclear what the bigger importance of this work is and if this has application to human health. Without human health connection, why does this matter to mice biology. "encoding Pro263, Val312, Pro330" means nothing. What is the sequence these refer to.

Results:

-Unclear how Table 1 shows anything about evolutionary selected mechanism. How was zygosity handled?

- non-synonymous amino acid changes is a redundant statement. These are the types of typos that make this paper a very tough read. This is just one example of the many writing issues I have.

-The phylogenetic tree in Figure S1 needs to be defined on what was used to create it in the results. If the allele is homozygous then this all seems obvious that they have to be related, so why waste the time in even talking about the tree outside of a simple statement of the variant

being conserved. This first paragraph is stretching out results way too much, which could be said in 2-3 simple sentences.

-“ no segregating non-synonymous structural variants (data not shown)”, with as much supplemental data presented, why is this not shown. In the era of open data and accessibility this is not okay to make statements like this.

-“ Given the evidence from inbred strains of mice indicating that a quarter or more of the mammalian genome consists of chromosomal regions containing clusters of functionally related genes”, I have no idea how this supports the claimed study. That just is not how evolution works. Yes genes can cluster, but unless more than 50% of the pathways is explained by the clustering, statistically the linkage means nothing to the assessment. Also how is this confirmed by inbred strains. That makes no sense.

General points:

1) The title is a bit confusing. BPHSE abbreviation is disconnected from the term and thus it takes looking to realize their connections or what the title means.

2) The abbreviations are very confusing when reading. They slow down the read and make it very difficult to follow the story without rereading and constantly going back to remind what each one is. It might not be the best use of abbreviation forms as they all bend together. Either writing these out or by changing them to the first letter being the variable instead of it being the last variables of the abbreviation. Bphse vs Bphs, it does not match well to have these so similar. For example if you are sold on abbreviations you could use BpHisS, BpHisSEnh BpSerS. These are more pronounced differences allowing the reader to use logic structure on each and thus read this quicker. Add on top of this the stain names and some sentences have more non English words than actual English words. For example, “We also found that Bphse requires Hrh1/HRH1, as no BC1 mice that genotype as Hrh1-/- were Bphss”

Reviewer #1

Item 1. I do wonder if the Bphse region is just picking up on a non-coding variant in a promoter or enhancer of *Hrh1*, or could be in a splice site mutation that reduces *Hrh1* expression. I think if the discussion was expanded to include this possibility, it would improve the manuscript.

Response: We have clarified (lines 224-237) that while this is a possibility because some of the strains are not deeply sequenced, it may not explain the susceptibility of strains where we do have deep sequenced data available and which exhibit no additional promoter/enhancer/non-coding variant of *Hrh1* that segregates with susceptibility.

Specific comments:

Item 2. The discussion on the evolutionary distribution of the two alleles using the mouse phylogenetic tree from Petkov et al 2004 may not be the most appropriate illustration. This is because the classical inbred strains have limited and non-randomly distributed genetic diversity, and the segregation of a particular locus may not follow the pattern of global genomic relatedness. Please review Yang et al. 2011 (<https://www.nature.com/articles/ng.847#Sec2>) as they illustrate the non-randomness of the mouse chromosome 6 in Figure 2. (I will attach this figure as well)

From the sequence analysis performed later, is it possible to reconstruct the relatedness between strains by origin or as haplotypes (i.e. *m. mus*, *m. cast*, *m. dom*) at this Bphse locus on chromosome 6, as done in the attached figure, and then highlight the bphs susceptibility?

Response: We agree that Chr 6 has regions of non-randomness that can impact the phylogenetic distribution (done by Petkov 2004); however, as we followed up the work mentioned in Yang et al., 2011 and manuscript (Figure 2), we noticed two issues that could impact this analysis:

- 1) There is heavy clustering of *M. m. musculus* and *M. m. castaneus* regions at Chr 6: 130-150 Mb among laboratory derived strains whereas the rest of the Chr has predominant contribution of *M. m. domesticus*. This could impact the construction of genetic relatedness based on which region is shortlisted for this work. Moreover, *Bphse* locus (Chr6:45.0-130.0Mb) excludes this region of variability in Chr 6 so any phylogenetic mapping done using sub-regions may not truly represent the distribution of whole mouse genomes. See Figure 1 (right):
- 2) Mapping of phylogenetic relatedness using the locus identified in this study (Chr6: 114780885-114758979) seems a plausible approach to derive phylogenetic relatedness however this may bias the contribution of co-segregating epistatic interactors of *Bphse*.

Fig 1. Chromosome 6 region is depicted with the cumulative contribution of *M. m. domesticus* (D, blue), *M. m. musculus* (M, red) and *M. m. castaneus* (C, green) subspecies. Source: <https://www.nature.com/articles/ng.847#Sec2>

Item 3. I see in figure 2B a band with 0 genetic variation between 114 and 115 that is likely one region with high identical-by-descent found in Yang et al. 2011. Interesting that *Hrh1* sits on just the edge of this region. **Could there be strain specific variation in this IBD region that was missed when you used the imputed genotypes?** This may be a concern as lab mouse strains are maintained with small breeding populations and can easily acquire and maintain novel mutations.

Response: There is a gap in the Mouse Diversity Array 114438359-114588610 (150kb) as highlighted in Yang et al., 2011 study that would explain limited coverage of this region. Also, this would preclude any Identity by Descent analysis for this region. This is a limitation in our study, and we have acknowledged it in the manuscript (lines 367-373):

“While every available resource was utilized to generate SNP data across Chr 6, there remain minute gaps in the coverage e.g. 114438359-114588610 (150kb) as apparent in Figure 2 and highlighted in Yang et al., 2011 study [67]. This would preclude any Identity by Descent analysis for this region and is a limitation in this study. As more mouse strains are sequenced in depth, future studies can clarify if there is a unique polymorphism in Chr6: 114438359-114588610 that may explain susceptibility to Bphs”

Item 4. You say that you know all the segregating nonsynonymous structural variants, **but could there be variants in promoter and enhancer regions affecting expression? How do you know that the Bphse locus, which contains *Hrh1*, isn't just picking up on non-coding variation?**

Response: We have updated the manuscript (lines 224-237) to reflect the claim based on available deep sequence data for the *Hrh1* gene between *Hrh1^r*/HRH1^r and *Hrh1^s*/HRH1^s mouse strains. There are only a few strains for which deep sequence data are not available so there is a possibility they may be carrying unidentified variants in promoter and enhancers of the *Hrh1* gene. However, that would not explain the susceptibility of respective members of Group 7 to Bphs (Supplementary Figure 1). Our claim is further supported by imputed SNP data generated across entire *Hrh1* and we didn't identify any additional nonsynonymous structural variant segregating between Bphse susceptible and resistant strains.

Item 5. **Have you looked at expression levels of *Hrh1* across available strains and compared it to your recorded resistant and sensitive phenotypes?** Yu et al. 2020 and Gu et al. 2020 (doi: 10.1111/epi.16741, doi: 10.1111/epi.16617) recently showed that a splice site variant in *Gabra2* (found in C57BL/6J!) reduced expression of this gene, leading to increased sensitivity to epilepsy. I really wonder if your SNP analysis is picking up on something like this. Can you rule this out in your study by looking at gene expression? I think you at least need to bring up this point in your discussion.

Response: We have not done this analysis, as we do not know the target cell type or tissue in which to measure expression, and we discuss this as a future direction. We have listed the possibility that HRH1 expression could be modified by Bphse in the discussion section (lines 386-388). The following has been added to the text (lines 396-7). “It will be informative to measure the surface expression of HRH1 among the *Hrh1^r*/HRH1^r strains that phenotype as Bphs^s (Table 2). Results using bone marrow chimeras suggest that Bphs^s is a function of the non-hematopoietic compartments [48], so several cell types (endothelial, epithelial, stromal cells) are potential candidates for this cell surface expression analysis”

Item 6. Page 22. “Control animals received carrier.” should be “A carrier”?

Response: We have updated to “animals received carrier alone” (line 450).

Item 7. Page 22 “DNA sequencing of third intracellular loop of Hrh1.” should be of THE third?

Response: We have made the change throughout.

Item 8. Page 24 “using Promega GoTaq according standard conditions“ should be corrected.

Response: We have made the change (line 484).

Item 9. Page 26 “The prepared library were then reacted with...” should be corrected.

Response: We have made the change.

Item 10. Page 30. Maybe this is an error with the pdf, but I don’t see a formula used for the combined genetic score.

Response: It is now visible in the word document (line 623). See also below:

$$S_{cg} = \frac{-\log_{10}(p_{EMMA})}{\max_{pos.cand.} -\log_{10}(p_{EMMA})} + \frac{-\log_{10}(FPR_{SVM})}{\max_{pos.cand.} -\log_{10}(FPR_{SVM})}$$

Item 11. Figure 2B, is it possible to display the gene names in a more organized way? Maybe it doesn’t matter so much if the same ones are represented in Fig 2C. If that is the case, just remove the names in 2B.

Response: We appreciate the suggestion. We have elected to show the gene names in Fig 2B to make it clear the location of *Hrh1* and other segregating variants with reference to the genomic position. We have, however, simplified the nomenclature throughout the text.

Item 12. Again, similar to Figure 3A. It is difficult to read the gene names here. They either need a smaller font, or should be organized differently because it is not easy to read.

Response: We have updated the figure to improve visualization of the gene names.

Item 13. Supplemental Figure 2 is impossible to read. Could this just be presented in a table?

Response: We have updated the figure to present a clear view of the three overlapping linkage disequilibrium domains underlying responsiveness to histamine.

Reviewer #2

Item 1. Abstract and other places – there is a terminology mix-up. Bphs is defined as a phenotype and not as a gene. Hence, using terms like Bphss/Bphsr alleles represents an oxymoron that needs to be sorted out.

Response: We agree that this can get confusing quickly, so we have given additional clarifications through insertion of “protein” or “phenotype” in key places and used consistent (italicization) and standard nomenclature of Bphs = phenotype; *Bphs* = gene/locus; BPHS = protein

Item 2. p. 5 - and other places. Terminology Hrh1r/HRH1r allele is used, where HRH1 is the gene product, protein, not the allele of Hrh1 gene. Correct and improve terminology, such shortcutting is unacceptable.

Response: *Hrh1* = gene/locus; HRH1 = protein. We have followed the MGI annotation guidelines for gene, protein, and phenotypes.

Item 3. p.7 - The authors sequenced 500 bp stretches of genomic DNA, whereas the HRH1 protein has 488 residues. Can the authors exclude that some Bphss strains genotyped as Hrh1r are in fact Hrh1s due to undiscovered polymorphisms outside of the sequenced 500 nt interval? Please, explain, elaborate on that in Result and in Discussion.

Response: We thank the reviewer for this comment and have complemented our Sanger sequencing data with publicly available sequenced data of mouse strains such as in Mouse Genomes Project and found no additional undiscovered polymorphisms outside of the sequenced 500 nt interval that segregated with susceptibility. Similarly, we performed targeted sequencing using Nimblegene across the entire *Hrh1* gene and again found no segregating polymorphism unaccounted for by our study. Last, we used SNP imputation to fill the gaps in some strains that were missing genotypes and again in agreement with the hypothesis found no additional segregating *Hrh1* polymorphism that would explain Bphs in Group 7 (Supplementary Figure 2). These approaches are highlighted in the methodology, and we also clarified in the Results section as suggested by reviewer #1.

Item 4. P.8 - comparison of the entire Hrh1 gene (Chr6:114,397,936-114,483,296 bp) between several of the Group 1 and Group 7 phenotyped strains found no segregating non-synonymous structural variants (data not shown). Does that mean that all phenotypically Bphs strains that were genotyped as Hrh1r, were included in sequencing of the entire genes? This should be specified.

Response: We have modified the statement (lines 148-153) to reflect the analysis we did. “Moreover, there were no segregating structural variants of the *Hrh1* gene (Chr6:114,397,936-114,483,296 bp) between representative Bphs^s and Bphs^r strains of phenotyped Group 1 and Group 7 mice. This is supported by imputed SNP datasets across all phenotyped strains (see methodology) suggesting that the complementing locus/loci in Group 7 is independent of previously unidentified *Hrh1*^s structural variants.”

Item 5. p. 8. HRH1KO is a non-standard designation of the Hrh1 knockout B6 strain. Its proper name should be at least once introduced to follow the rules of the International Committee on Standardized Genetic Nomenclature for Mice. It would also help to show a scheme with all the used combinations of Hrh1 and Bphs-e genotypes of the strains used in the backcrosses.

Response: We have included the following clarification on line 157 of the revised manuscript “*Hrh1*-knockout B6 mice (B6.129P2-*Hrh1*^{tm1Wtn}/BrenJ, common name H1R KO)” and then consistently use H1R KO thereafter.

Item 6. P.8 Is there any phenotype difference between Bphs sensitivity of HRH1KO/KO and *Hrh1*s/*Hrh1*s?

Response: In lines 158-9 we have clarified that mouse strains that do not express a functional *Hrh1*^s gene (e.g. HRH1KO/KO or *Hrh1*^l) do not exhibit Bphs.

Item 7. P.9. Six out of 60 of offspring of (MOLF × HRH1KO) × HRH1KO backcross were apparent recombinants between the *Hrh1* gene and Bphs-e locus suggesting 10cM or cca 20 Mb distance between them. But such conclusion would contradict the colocalization of *Hrh1* and Bphs within a 5 Mb interval. Please explain/discuss.

Response: We agree. The number of recombinants found are few and would benefit if more F2 offspring were studied. This number could change accordingly and hence we do not claim a 5Mb interval distance based on the recombination data. More animals need to be studied to translate recombination rates to linkage. We do note however in the text that the interval between D6Mit102 and rs31698248 is 26 MB (line 194).

Item 8. The authors need to acknowledge that the found 10 candidates in the 5 Mb segment is not an exhaustive list of potential modifiers (enhancers, non-coding RNAs, gene models with unknown function, etc.)

Response: We fully agree and we have added to text (lines 410-415). “Given that this study has shortlisted candidates for Bphse to 10 candidates, it is important to highlight that this is not an exhaustive list of potential modifiers (enhancers, non-coding RNAs, genes with unknown function, etc.) and follow up studies are needed to determine the causal loci for Bphse. One way to test this would be to quantify the mRNA expression of some of these top candidates between Bphs^s and Bphs^l strains and investigate whether they interact with HRH1”

Reviewer #3

Item 1. Abstract: In general the abstract uses too many vague descriptions and is very diffuse to what the actual results are, making it a bit frustrating to read, such that one needs to go to the full paper to have any idea what is actually being done. As the work is strong, the abstract needs a nearly complete rewrite. Need to define the subset of mice used to show susceptibility. What is the exact allele driving change? The “extended panel” needs to be defined. Exactly how many of them. Are these a traditional panel? The “eight strains” is vague, are these 8 related strains and thus could be referred to instead based on the evolutionary divergence points? “Genetic analyses” is too vague. What does that mean. Narrowing to a 5.5 Mb region does not seem publishable as a major discovery to end the abstract on. The end of the abstract leaves more work than what the start of the abstract set out to do. This loses me and detracts from the awesome work in the paper itself. It makes more sense to frame this paper as setting out to discover if other alleles outside those known can impact histamine sensitization, where the conclusion is a resounding yes. That does not leave the story hanging for so much follow up validation.

Response: We agree and we have edited the abstract as below.

Histamine plays pivotal role in normal physiology and dysregulated production of histamine or signaling through histamine receptors (HRH) can promote pathology.

Previously, we showed that *Bordetella pertussis* or pertussis toxin can induce histamine sensitization in laboratory inbred mice and is genetically controlled by *Hrh1*/HRH1. HRH1 allotypes differ at three amino acid residues with P₂₆₃-V₃₁₃-L₃₃₁ and L₂₆₃-M₃₁₃-S₃₃₁, imparting sensitization and resistance respectively. Unexpectedly, we found several wild-derived inbred strains that carry the resistant HRH1 allotype (L₂₆₃-M₃₁₃-S₃₃₁) but exhibit histamine sensitization. This suggests the existence of a locus modifying pertussis-dependent histamine sensitization. Congenic mapping identified the location of this modifier locus on mouse chromosome 6 within a functional linkage disequilibrium domain encoding multiple loci controlling sensitization to histamine. We utilized interval-specific single-nucleotide polymorphism (SNP) based association testing across laboratory and wild-derived inbred mouse strains and functional prioritization analyses to identify candidate genes for this modifier locus. *Atg7*, *Plxnd1*, *Tmcc1*, *Mkrn2*, *Il17re*, *Pparg*, *Lhfp14*, *Vgll4*, *Rho* and *Syn2* are candidate genes within this novel modifier locus, which we named *Bphse*, enhancer of *Bordetella pertussis* induced histamine sensitization. Taken together, these results identify, using the evolutionarily significant diversity of wild-derived inbred mice, novel genetic mechanisms controlling histamine sensitization..

Item 2. Introduction: The introduction is filled with poor writing and grammatical issues. It is very tough to read. This is so focused on the mouse that it is very unclear what the bigger importance of this work is and if this has application to human health. Without human health connection, why does this matter to mice biology.

Response: We thank the reviewer for this input and we have updated the introduction with relevance of HA to human physiology emphasized (lines 64-69), and undertook additional proof-reading to correct grammatical issues.

Item 3. “encoding Pro263, Val312, Pro330” means nothing. What is the sequence these refer to?

Response: We have clarified (lines 95-98) that these refer to “amino acids”.

Item 4. Results: Unclear how Table 1 shows anything about evolutionary selected mechanism. How was zygosity handled?

Response: We don't claim Table 1 shows the evolutionary selected mechanism. We used Supplementary Figure 1 to make that claim and it is based on the distribution of *Hrh1* haplotypes and Bphs phenotype.

Item 5. - non-synonymous amino acid changes is a redundant statement. These are the types of typos that make this paper a very tough read. This is just one example of the many writing issues I have.

Response: The reviewer is correct and we have now used “amino acid changes” (e.g. line 120-123). We have restricted “synonymous/non-synonymous” usage to genetic sequence (e.g. Table S2).

Item 6. The phylogenetic tree in Figure S1 needs to be defined on what was used to create it in the results. If the allele is homozygous then this all seems obvious that they have to be related, so why waste the time in even talking about the tree outside of a simple statement of the variant being conserved. This first paragraph is stretching out results way too much, which could be said in 2-3 simple sentences.

Response: The phylogenetic tree is presented in Petkov et al., 2004 and we utilized it to show the distribution of Bphs phenotype and segregation with *Hrh1* haplotype. We have

condensed our conclusion to a single sentence (lines 126-130). We retained this because we feel it is important to explain the genetic makeup of Group 7 to explain the point that the haplotype complementing sensitization to histamine is independent of additional previously unidentified *Hrh1^s* structural variants (lines 143-153).

Item 7. “ no segregating non-synonymous structural variants (data not shown)”, with as much supplemental data presented, why is this not shown. In the era of open data and accessibility this is not okay to make statements like this.

Response: We appreciate the reviewer’s sentiment and have reworded this statement (lines 147-150) and removed reference to “data not shown”. Furthermore we explain our imputation methods which support our conclusion that no Bphs-segregating structural variants of the *Hrh1* gene were found across Group 1 and Group 7 strains. A table could be generated for the supplemental information but we feel that this is unnecessary and could be independently validated using existing sequencing data.

Item 8. “ Given the evidence from inbred strains of mice indicating that a quarter or more of the mammalian genome consists of chromosomal regions containing clusters of functionally related genes”, I have no idea how this supports the claimed study. That just is not how evolution works. Yes genes can cluster, but unless more than 50% of the pathways is explained by the clustering, statistically the linkage means nothing to the assessment. Also how is this confirmed by inbred strains. That makes no sense.

Response: Our hypothesis is that the locus surrounding *Hrh1* represents a functional LD domain (a concept that is supported by the citations provided) containing a cluster of genes regulating histamine responsiveness. In this study, we identified Bphse in linkage with Bphs (identified in Ma et al., 2002 as *Hrh1*) and Histh (identified in Raza et al., 2019), all regulating histamine responsiveness in mice. This is elaborated in Supplementary Figure 2. Inbred strains simply provide a wide selection of genotypes to identify genes variants in this locus.

General points:

1) The title is a bit confusing. BPHSE abbreviation is disconnected from the term and thus it takes looking to realize their connections or what the title means.

Response: We agree and the title was edited as follows “A NOVEL LOCUS COMPLEMENTS RESISTANCE TO BORDETELLA PERTUSSIS-INDUCED HISTAMINE SENSITIZATION”

2) The abbreviations are very confusing when reading. They slow down the read and make it very difficult to follow the story without rereading and constantly going back to remind what each one is. It might not be the best use of abbreviation forms as they all bend together. Either writing these out or by changing them to the first letter being the variable instead of it being the last variables of the abbreviation. Bphse vs Bphs, it does not match well to have these so similar. For example if you are sold on abbreviations you could use BpHisS, BpHisSEnh BpSerS. These are more pronounced differences allowing the reader to use logic structure on each and thus read this quicker. Add on top of this the stain names and some sentences have more non English words than actual English words. For example, “We also found that Bphse requires Hrh1/HRH1, as no BC1 mice that genotype as Hrh1^{-/-} were Bphss”

Response: We apologize for the inconsistent nomenclature and we have revised several parts of the Abstract and the manuscript to reduce the confusion with nomenclature. To clarify we

are utilizing the MGI Nomenclature classification scheme: Bphs = phenotype; *Bphs*, *Bphse*, *Hisths*, *Hrh1* = gene/locus; *Hrh1^s* = susceptible haplotype; *Hrh1^r* = resistant haplotype; HRH1 = protein

REVIEWERS' COMMENTS:

Reviewer #1 (Remarks to the Author):

The comments have been adequately addressed by the authors in the revised manuscript.

Reviewer #3 (Remarks to the Author):

The majority of comments were successfully addressed. Belows is only one remaining minor concerns needing corrected. Great job!

Item 3. "encoding Pro263, Val312, Pro330" means nothing. What is the sequence these refer to? Before listing the changes, you must list the accession code for the exact sequence, so anyone can be 100% confident which amino acid this is referring to.

Reviewer #3 (Remarks to the Author):

The majority of comments were successfully addressed. Below is only one remaining minor concerns needing corrected. Great job!

Item 3. “encoding Pro263, Val312, Pro330” means nothing. What is the sequence these refer to? Before listing the changes, you must list the accession code for the exact sequence, so anyone can be 100% confident which amino acid this is referring to.

Response: We have provided the accession code for HRH1 sequence and also added in-text the total number of amino acids in HRH1 protein.

“HRH1 encodes a protein with 488 amino acids (UniProtKB identifiers: P70174; www.uniprot.org). Susceptibility to Bphs segregates with two conserved *Hrh1* haplotypes, mice with the *Hrh1^s* allele (encoding HRH1^s amino acids Pro, Val, Pro at position 263, 312 and 330 respectively in the primary sequence of HRH1 protein) are sensitive to PTX (Bphs^s) while mice with the *Hrh1^r* allele (encoding HRH1^r amino acids Leu, Met, Ser at position 263, 312 and 330) are resistant to PTX (Bphs^r).”